# ADAPTIVE PROJECTED GUIDANCE FOR CONTROLLABLE INSTRUCTION-GUIDED IMAGE EDITING

## ABSTRACT

Instruction-guided diffusion models have demonstrated strong capabilities in generating targeted image edits based on diverse textual prompts. A fundamental challenge in this setting is achieving the right balance between adhering to textual instructions and preserving the original content of the input image. Instruct-Pix2Pix (IP2P) addresses this by applying separate classifier-free guidance (CFG) terms to the text and image conditions, each scaled independently. However, this limited parametrization restricts user control, as increasing one guidance scale often causes the corresponding condition to dominate the output, resulting in imbalanced edits. Independently, Adaptive Projected Guidance (APG) was recently introduced to mitigate inherit limitations of CFG at high guidance scales in text- and class-conditioned diffusion models, reframing CFG as a gradient ascent process with decomposed guidance directions and improved signal control. In this work, we present IP2P-APG, a plug-and-play extension of IP2P that repurposes APG to improve the balance between instruction adherence and content preservation in image editing tasks. IP2P-APG significantly expands the controllable parameter space, allowing users to have more precise control over the editing process. Moreover, by enabling the use of higher guidance scales without introducing artifacts or compromising fidelity to the original content, IP2P-APG achieves a more effective trade-off between textual alignment and content preservation. Extensive experiments across multiple generative backbones and datasets demonstrate that our method consistently produces more realistic and instruction-faithful edits, without additional training and with negligible computational overhead. Code will be released after the review process.

## 1 INTRODUCTION

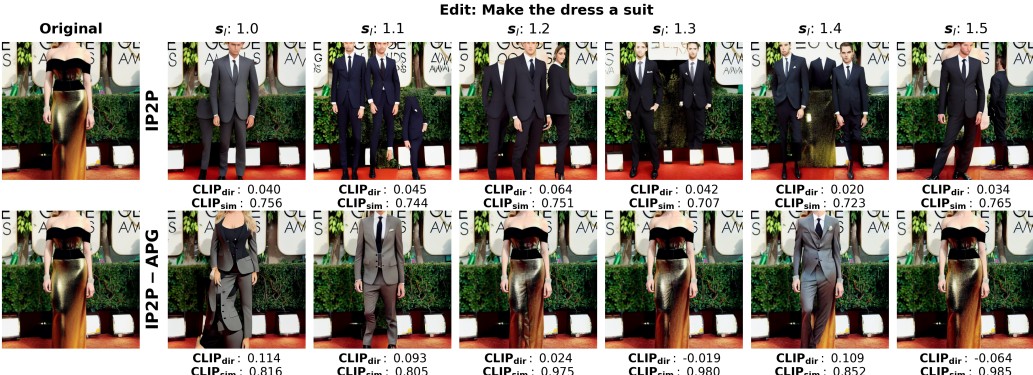

Figure 1: Comparison of IP2P (Brooks et al., 2023) (first row) and IP2P-APG (second row) with $s_T = 7.5$ while varying $s_I$. Here $s_T$ is the scaling factor on text guidance and $s_I$ is the scaling factor on image guidance during the editing. IP2P-APG maintains the subject's posture and orientation more consistently while using the same model parameters as IP2P, and with $s_I = 1.4$ it achieves a strong trade off between edit strength and fidelity.

Instruction-guided image editing has emerged as a powerful approach for enabling everyday users to modify images by simply describing the desired changes in natural language. This technique has received considerable attention from the research community, prompting the creation of numerous datasets and methods (Brooks et al., 2023; Zhang et al., 2023; Sheynin et al., 2024; Zhao et al., 2025; Hui et al., 2024) that seek to expand the capabilities and accessibility of instruction-based editing. Among these advancements, InstructPix2Pix (IP2P) (Brooks et al., 2023) stands out for its novel framework that uses text-conditional, image-conditional, and unconditional noise estimations.

A core challenge in this line of work is maintaining a balance between faithfully applying the requested transformations and preserving the intrinsic fidelity of the input image. IP2P tries to address this by introducing separate scale factors for its text- and image-conditioned noise estimations, thereby affording control over each aspect of the edit. While increasing these scales can help to achieve stronger text-driven modifications or reinforce image fidelity, pushing them undermine the overall content quality. Specifically, an excessively large text scale overwrites important details of the source, whereas a disproportionately high image scale suppresses the desired edits, leaving the image nearly unchanged. In practice, identifying the optimal guidance scales demands extensive empirical tuning, offering very limited control over the final output. The inherent constraints in adjusting these scales mean that users are left with only a narrow margin for tailoring the balance between the textual instruction and the preservation of the original image. Even minor miscalibrations can result in outputs where the textual directive is either underrepresented or the source content is excessively distorted. This pronounced sensitivity underscores a broader challenge in instruction-guided diffusion models: the limited user control available to achieve a harmonious balance between faithfully incorporating the requested change and maintaining the essential attributes of the original image.

These fundamental limitations point to a need for more sophisticated control mechanisms. Fortunately, recent advances in text-to-image diffusion models offer promising directions. Specifically, studies have shown that while high classifier-free guidance (CFG) scales can enhance generation quality, they often lead to oversaturation and unrealistic artifacts in generated images (Sadat et al., 2025). Adaptive Projected Guidance (APG) addresses these challenges by decomposing the CFG update into parallel and orthogonal components relative to the conditional estimate (Sadat et al., 2025). Through strategic rescaling and reverse momentum, APG effectively controls the oversaturation while preserving the benefits of CFG (Sadat et al., 2025). These insights provide a foundation for addressing the control limitations in instruction-guided image editing.

Motivated by these findings, we propose **IP2P-APG**, a plug-and-play training-free framework that extends adaptive projection strategies for instruction-guided image editing. Our approach systematically addresses the control limitations of existing methods by decomposing the guidance signals into parallel and orthogonal components, introducing separate momentum and normalization terms for text and image guidance signals, hence enabling more nuanced control of the editing process. This principled decomposition allows IP2P-APG to move beyond the rigid coupling of guidance scales, achieving both faithful content preservation and precise alignment with user instructions. IP2P-APG is model-agnostic and demonstrates effectiveness across diverse instruction-guided image editing backbones, from U-Net-based latent diffusion models (Rombach et al., 2022) to rectified flow transformers (Esser et al., 2024). This versatility is demonstrated by integrating IP2P-APG into three state-of-the-art instruction-guided editors: InstructPix2Pix (Brooks et al., 2023), MagicBrush (Zhang et al., 2023), and UltraEdit (Zhao et al., 2025).

Our contributions are listed as follows:

- We develop a plug-and-play framework for instruction-guided image editing that expands user control by parameterizing the editing dynamics along three axes: momentum, normalization, and parallel components. This provides interpretable levers for calibrating edit intensity, mitigating artifacts, and preserving desired source attributes.

- We provide a systematic analysis of how these controls affect the denoising dynamics under joint text and image guidance, revealing how users can trade off edit strength, fidelity, and content preservation.

- Our model-agnostic framework achieves stronger content fidelity and instruction alignment with minimal overhead, as shown across benchmarks with diverse generative models.

## 2 METHODOLOGY

We begin this section by establishing the theoretical foundations of our work, covering classifier-free guidance, instruction-guided image editing, and adaptive projected guidance. Following this background review, we present our proposed methodology in detail.

### 2.1 CLASSIFIER-FREE GUIDANCE (CFG)

CFG is an inference method designed to enhance the quality of generated outputs by combining the predictions of a conditional model and an unconditional model (Ho & Salimans, 2022). Given a null condition $c_{\text{null}} = \varnothing$ for the unconditional case, CFG modifies the denoiser's output at each sampling step as follows:

$$\tilde{\epsilon}_\theta\left(z_t, c\right) = \epsilon_\theta\left(z_t, \varnothing\right) + w \cdot \left(\epsilon_\theta\left(z_t, c\right) - \epsilon_\theta\left(z_t, \varnothing\right)\right), \tag{1}$$

where $w = 1$ represents the non-guided case, $z_t$ the noisy sample at time index t, and $\theta$ denoiser network parameters. The unconditional model $\epsilon_\theta(z_t, \varnothing)$ is trained by randomly applying the null condition $c_{\text{null}} = \varnothing$ to the denoiser's input for a portion of training.

### 2.2 ADAPTIVE PROJECTED GUIDANCE (APG)

APG has been recently introduced as a solution that preserves the advantages of CFG while significantly reducing artifacts that typically emerge at higher guidance scales (Sadat et al., 2025). APG operates on denoised predictions (predicted clean latent, denoted as $D(.)$). For consistency, we maintain this notation throughout our analysis. The standard classifier-free guidance update with a given noisy sample $\mathbf{z}_t$, condition $\mathbf{c}$, and guidance scale $w$ is given by:

$$\tilde{D}_\theta(\mathbf{z}_t, \mathbf{c}) = D_\theta(\mathbf{z}_t, \mathbf{c}) + (w - 1) \Delta D_t, \tag{2}$$

where $\Delta D_t$ represents the difference between conditional and unconditional (null condition, denoted as $\varnothing$) estimates of the denoised image:

$$\Delta D_t = D_\theta(\mathbf{z}_t, \mathbf{c}) - D_\theta(\mathbf{z}_t, \varnothing),$$

While increasing $w$ generally improves sample fidelity and condition alignment, it can also lead to oversaturation and visual artifacts.

CFG can also be interpreted as performing a single step of gradient ascent on the squared difference between conditional and unconditional predictions (Sadat et al., 2025):

$$\frac{1}{2} \left\| D_\theta(\mathbf{z}_t, \mathbf{c}) - D_\theta(\mathbf{z}_t, \varnothing) \right\|^2. \tag{3}$$

From this viewpoint, $\Delta D_t$ in Eq. (2) acts much like a gradient step with an effective "learning rate" of $(w - 1)$. When $w$ grows large, however, this update can overshoot, pushing the model beyond a desirable range. To counter this, APG introduces rescaling and momentum, techniques borrowed from optimization, to keep the updates in check.

**Parallel and Orthogonal Decomposition:** To better understand which aspects of $\Delta D_t$ contribute to oversaturation, APG decomposes it into parallel and orthogonal components relative to $D_\theta(\mathbf{z}_t, \mathbf{c})$. The parallel component is given by:

$$\Delta D_t^\| = \frac{\langle \Delta D_t, \, D_\theta(\mathbf{z}_t, \mathbf{c}) \rangle}{\langle D_\theta(\mathbf{z}_t, \mathbf{c}), \, D_\theta(\mathbf{z}_t, \mathbf{c}) \rangle} D_\theta(\mathbf{z}_t, \mathbf{c}), \tag{4}$$

which leads to the decomposition:

$$\Delta D_t^\perp = \Delta D_t - \Delta D_t^\|. \tag{5}$$

Empirical observations indicate that the parallel component $\Delta D_t^\|$ is primarily responsible for increasing pixel intensities, which can result in oversaturation. Conversely, the orthogonal component $\Delta D_t^\perp$ plays a more significant role in improving image quality (Sadat et al., 2025).

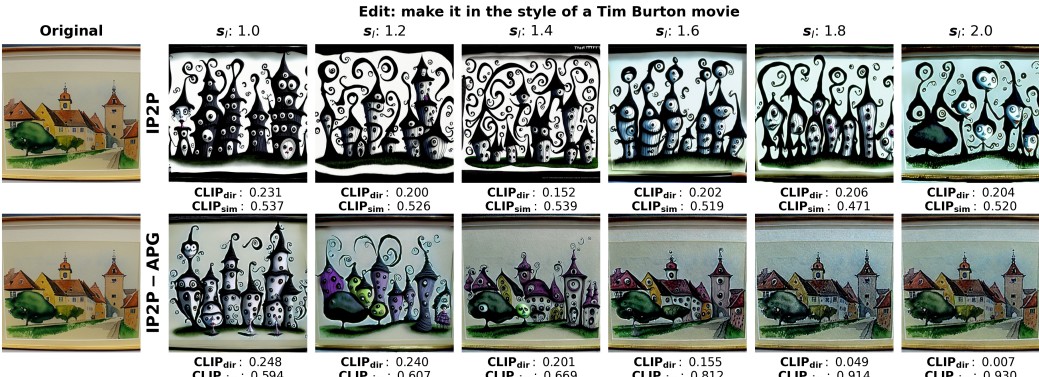

Figure 2: Comparison of IP2P (Brooks et al., 2023) and IP2P-APG at $s_T = 15.0$ with varying $s_I$ values. Despite the high $s_T$ value, IP2P-APG accurately preserves the input structure, whereas IP2P fails to do so.

**Rescaling and Reverse Momentum:** To regulate the update magnitude, APG applies rescaling to $\Delta D_t$, ensuring that it remains within an $\ell_2$-ball of radius $r$:

$$\Delta D_t \leftarrow \Delta D_t \cdot \min\left(1, \; \frac{r}{\|\Delta D_t\|}\right). \tag{6}$$

This prevents excessively large updates and limits unnatural deviations from $D_\theta(\mathbf{z}_t, \mathbf{c})$.

Additionally, APG incorporates a reverse momentum mechanism to counteract excessive accumulation of updates in the same direction (Sadat et al., 2025). Specifically, a negative momentum factor $\beta < 0$ is introduced:

$$\overline{\Delta D_t} \leftarrow \Delta D_t + \beta \, \overline{\Delta D_t}, \tag{7}$$

where $\overline{\Delta D_t}$ maintains a running average of past updates.

Unlike standard momentum, which reinforces past update directions, this repulsive momentum ensures that each new update is adjusted away from previous directions, helping to suppress excessive accumulation in the parallel component.

## 2.3 INSTRUCTPIX2PIX (IP2P)

Traditional single-condition CFGs usually focus on a single modality, typically text. However, instruction-guided image editing requires additional control to preserve the integrity of the input image. IP2P addresses this by employing distinct guidance parameters for the image and text conditions, denoted by $s_I$ and $s_T$, respectively. Specifically, $s_I$ governs how strictly the output should adhere to the original image's structure (e.g., color palette, composition), preventing large deviations when $s_I$ is high. Meanwhile, $s_T$ dictates how strongly the model should follow the textual instruction, allowing more creative or substantial modifications if set to a higher value.

Hence, the modified noise estimate at step $t$ in IP2P is expressed as:

$$\begin{aligned}
\tilde{e}_\theta\left(z_t, c_I, c_T\right) = &\, e_\theta\left(z_t, \varnothing, \varnothing\right) \\
&+ s_I \cdot \left(e_\theta\left(z_t, c_I, \varnothing\right) - e_\theta\left(z_t, \varnothing, \varnothing\right)\right) \\
&+ s_T \cdot \left(e_\theta\left(z_t, c_I, c_T\right) - e_\theta\left(z_t, c_I, \varnothing\right)\right),
\end{aligned} \tag{8}$$

where $c_I$ stands for the image condition, $c_T$ for the text condition (i.e., instruction), and $\varnothing$ for the null condition.

## 2.4 THE IP2P-APG FRAMEWORK

In our proposed method, we rewrite the Eq. (8) with denoised predictions instead by substituting

$$e_\theta(z_t, c) = \frac{z_t - \alpha_T * D_\theta(\mathbf{z}_t, \mathbf{c})}{\sigma_T}, \tag{9}$$

where $z_t = \alpha_T x_0 + \sigma_T \epsilon$ follows the standard DDPM noise schedule (Ho et al., 2020). For rectified flow models like SD3 (Esser et al., 2024), we apply an analogous transformation to convert velocity predictions to denoised estimates. Consequently, the conditional denoising process can be expressed as:

$$
\begin{aligned}
\tilde{D}_\theta\left(z_t, c_I, c_T\right) = & D_\theta\left(z_t, \varnothing, \varnothing\right) \\
& + s_I \cdot \left(D_\theta\left(z_t, c_I, \varnothing\right) - D_\theta\left(z_t, \varnothing, \varnothing\right)\right) \\
& + s_T \cdot \left(D_\theta\left(z_t, c_I, c_T\right) - D_\theta\left(z_t, c_I, \varnothing\right)\right).
\end{aligned}
\tag{10}
$$

By defining the difference terms explicitly, we rewrite the equation as:

$$
\begin{aligned}
\tilde{D}_\theta\left(z_t, c_I, c_T\right) = & D_\theta\left(z_t, \varnothing, \varnothing\right) \\
& + s_I \cdot \Delta D_t^{I,\varnothing} \\
& + s_T \cdot \Delta D_t^{T,I},
\end{aligned}
\tag{11}
$$

where the differences are given by:

$$
\begin{aligned}
\Delta D_t^{I,\varnothing} &= D_\theta\left(z_t, c_I, \varnothing\right) - D_\theta\left(z_t, \varnothing, \varnothing\right), \\
\Delta D_t^{T,I} &= D_\theta\left(z_t, c_I, c_T\right) - D_\theta\left(z_t, c_I, \varnothing\right).
\end{aligned}
\tag{12}
$$

Observing that the term $D_\theta(z_t, \varnothing, \varnothing) + s_I \cdot \Delta D_t^{I,\varnothing}$ follows the same structure as Eq. (2), we can express Eq. (11) in a similar form by adding $D_\theta(z_t, c_I, \varnothing)$ to both sides. This yields:

$$
\begin{aligned}
\tilde{D}_\theta\left(z_t, c_I, c_T\right) = & D_\theta\left(z_t, c_I, \varnothing\right) + (s_I - 1) \cdot \Delta D_t^{I,\varnothing} \\
& + D_\theta(z_t, c_I, c_T) + (s_T - 1) \cdot \Delta D_t^{T,I} \\
& - D_\theta(z_t, c_I, \varnothing).
\end{aligned}
\tag{13}
$$

Finally, we apply the concepts of rescaling, momentum, and orthogonality to the difference terms $\Delta D_t^{I,\varnothing}$ and $\Delta D_t^{T,I}$, which enables precise control over text and image guidance through distinct momentum, orthogonality, and normalization hyperparameters.

The final IP2P-APG denoised prediction is given by

$$
\begin{aligned}
\tilde{D}_\theta(z_t, c_I, c_T) = & D_\theta(z_t, c_I, c_T) \\
& + (s_I - 1)\left\{M_t^{I,\varnothing} \min\left(1, \frac{r_I}{\|M_t^{I,\varnothing}\|}\right)\right\}_{D_\theta(z_t, c_I, \varnothing)}^{\perp} \\
& + (s_T - 1)\left\{M_t^{T,I} \min\left(1, \frac{r_T}{\|M_t^{T,I}\|}\right)\right\}_{D_\theta(z_t, c_I, c_T)}^{\perp},
\end{aligned}
\tag{14}
$$

where $r_I$ and $r_T$ are normalization radius for L2 norm, $M_t^{I,\varnothing}$ and $M_t^{T,I}$ are momentum-applied differences defined as

$$
\begin{aligned}
M_t^{I,\varnothing} &= \Delta D_t^{I,\varnothing} + \beta_I \overline{\Delta D}_t^{I,\varnothing}, \\
M_t^{T,I} &= \Delta D_t^{T,I} + \beta_T \overline{\Delta D}_t^{T,I},
\end{aligned}
$$

where $\beta_I$ and $\beta_T$ denote the momentum coefficients, and $\overline{\Delta D}_t^{I,\varnothing}$ and $\overline{\Delta D}_t^{T,I}$ are the running averages of the corresponding updates.

In this notation, we define the contribution of the parallel component using the orthogonal projection of a vector $X$ onto a vector $D$ as:

$$
\{X\}_D^{\perp} = X - (1 - \eta)\frac{\langle X, D \rangle}{\langle D, D \rangle}D,
$$

where the parameter $\eta$ controls the influence of the parallel component in the final estimate. Specifically, we use $\eta_I$ and $\eta_T$ to regulate the strength of the parallel components for $\Delta D_t^{I,\varnothing}$ and $\Delta D_t^{T,I}$, respectively.

Table 1: Comparison of methods in MagicBrush (Zhang et al., 2023) test set. For IP2P-APG, the checkpoint used is indicated in the first column. Best results are highlighted in bold.

| Method | L1 ($\downarrow$) | DINO ($\uparrow$) | CLIP$_I$ ($\uparrow$) | CLIP$_T$ ($\uparrow$) | CLIP$_{sim}$ ($\uparrow$) | CLIP$_{dir}$ ($\uparrow$) |
|---|---|---|---|---|---|---|
| InstructPix2Pix (Brooks et al., 2023) | 0.1132 | 0.7409 | 0.8534 | 0.2757 | 0.8069 | 0.1117 |
| MagicBrush (Zhang et al., 2023) | 0.0748 | 0.8475 | 0.9076 | 0.2848 | 0.8471 | **0.1380** |
| EmuEdit* (Sheynin et al., 2024) | - | - | - | - | **0.8970** | 0.1350 |
| UltraEdit (Zhao et al., 2025) | 0.0687 | 0.8450 | 0.8997 | 0.2869 | 0.8374 | 0.1261 |
| IP2P-APG (InstructPix2Pix SD1.5) | 0.1028 | 0.8026 | 0.8774 | 0.2849 | 0.8154 | 0.1287 |
| IP2P-APG (MagicBrush SD1.5) | 0.0667 | **0.8754** | **0.9175** | 0.2860 | 0.8556 | 0.1367 |
| IP2P-APG (Ultraedit SD3) | **0.0668** | 0.8424 | 0.9002 | **0.2892** | 0.8318 | 0.1366 |

*Since Emuedit does not provide metrics using ground truth images, we exclude those specific metrics from table (see Subsection 3.2 for details).

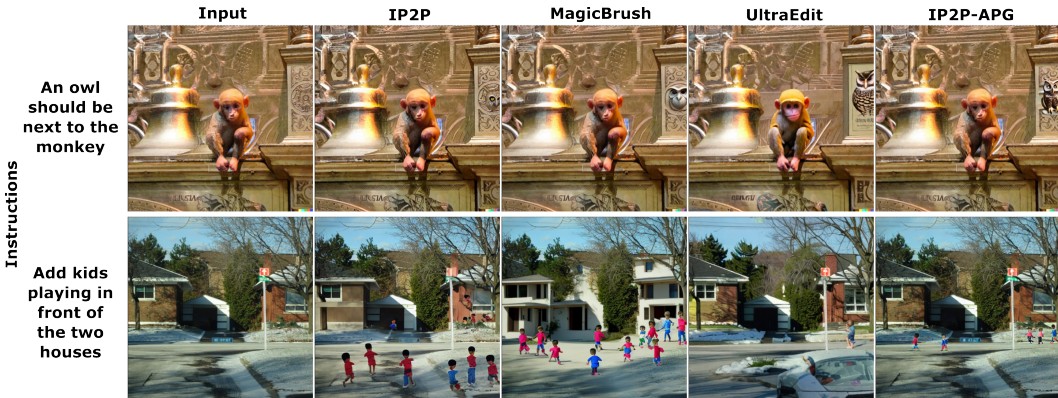

Figure 3: Comparison of methods for $(s_T, s_I) = (7.5, 1.5)$ using IP2P-APG with the MagicBrush (Zhang et al., 2023) checkpoint using MagicBrush test set. In the first row, IP2P-APG preserves the details of the monkey while applying edits accurately, while in the second row, it maintains the landscape details while ensuring precise modifications.

## 3 RESULTS

### 3.1 EXPERIMENTS

We compare our method against several baselines to evaluate its effectiveness. Specifically, we benchmark our approach against the original InstructPix2Pix (IP2P) (Brooks et al., 2023) using its fine-tuned checkpoint from Stable Diffusion v1.5, MagicBrush (Zhang et al., 2023) fine-tuned on its dataset with the latest Stable Diffusion v1.5 checkpoint from the MagicBrush repository, and UltraEdit (Zhao et al., 2025), which employs a Stable Diffusion 3-based checkpoint fine-tuned on its dataset. For EmuEdit (Sheynin et al., 2024), we utilize the provided generated results for the EmuEdit test set, as its codebase and model checkpoints are not publicly available.

Our evaluations are conducted on three distinct test sets: the InstructPix2Pix test set (15,651 images), the MagicBrush test set (1,056 images), and the EmuEdit test set (3,589 images). To assess performance trade-offs, we present comparative graphs illustrating the relationship between CLIP$_{sim}$ and CLIP$_{dir}$ metrics (see next section for detailed descriptions). These graphs are shown for the InstructPix2Pix test set in Appendix Fig. B.1, the MagicBrush test set in Appendix Fig. B.2, and the EmuEdit test set in Appendix Fig. B.3. Additionally, we provide quantitative results for the MagicBrush and EmuEdit datasets in Tabs. 1 and 2, respectively.

Furthermore, qualitative comparisons between IP2P and our proposed IP2P-APG method are presented in Fig. 1, Fig. 2, Appendix Fig. C.2 and Appendix Fig. C.4. Comparisons on the MagicBrush dataset are shown in Fig. 3 and on the EmuEdit dataset in Fig. 5. As shown in Fig. 1, our method significantly improves input structure preservation and textual adherence compared to the original

Table 2: Comparison of methods in Emuedit (Sheynin et al., 2024) test set. For IP2P-APG, the checkpoint used is indicated in the first column. Best results are highlighted in bold.

| Method | L1 ($\downarrow$) | DINO ($\uparrow$) | CLIP$_T$ ($\uparrow$) | CLIP$_{sim}$ ($\uparrow$) | CLIP$_{dir}$ ($\uparrow$) |
|---|---|---|---|---|---|
| InstructPix2Pix (Brooks et al., 2023) | 0.1220 | 0.7615 | 0.2541 | 0.8495 | 0.0686 |
| MagicBrush (Zhang et al., 2023) | 0.0826 | 0.8093 | 0.2535 | 0.8775 | 0.0853 |
| EmuEdit (Sheynin et al., 2024) | 0.0890 | 0.8398 | 0.2540 | 0.8743 | **0.1090** |
| Ultraedit (Zhao et al., 2025) | 0.0549 | 0.8470 | 0.2504 | 0.8757 | 0.0865 |
| IP2P-APG (InstructPix2Pix SD1.5) | 0.1069 | 0.8044 | **0.2573** | 0.8626 | 0.0818 |
| IP2P-APG (MagicBrush SD1.5) | 0.0574 | **0.8931** | 0.2490 | **0.9171** | 0.0823 |
| IP2P-APG (Ultraedit SD3) | **0.0525** | 0.8473 | 0.2518 | 0.8754 | 0.0949 |

IP2P. In Fig. 2, we illustrate the advantages of IP2P-APG under a higher textual guidance scale ($s_T = 15.0$), highlighting its effectiveness at larger scales, still preserving the input structure while delivering stronger edits. In the first row of Fig. 3, IP2P-APG accurately adds the owl while keeping the monkey's facial details intact; in the second row, it preserves landscape details better than the competing methods. As shown in Fig. 5, results with the SD3 backbone from UltraEdit demonstrate the versatility of our framework. In this setting, our method attains performance comparable to EmuEdit without access to its private training data.

## 3.2 COMPARISON METRICS

Building on existing literature (Brooks et al., 2023; Sheynin et al., 2024; Zhao et al., 2025), we assess image fidelity using multiple metrics. First, we compute the pixel-level L1 difference between the ground truth and edited images to quantify the preservation of original details. We also extract DINO features from both the original and edited images and compute their cosine similarity. Additionally, we evaluate CLIP image similarity (CLIP$_I$) by measuring the cosine similarity between the edited image and the ground truth. Finally, we calculate the cosine similarity between the input and edited image embeddings (CLIP$_{sim}$). To assess how well the image editing aligns with the provided text prompt, we incorporate two CLIP-based metrics: CLIP$_{dir}$ and CLIP$_T$. CLIP direction (CLIP$_{dir}$) evaluates alignment by comparing the transformation from the input to the output—both in terms of images and captions—with their corresponding descriptive captions (Brooks et al., 2023). Meanwhile, CLIP-text (CLIP$_T$) quantifies the similarity between the editing prompt and the edited image. For MagicBrush (Zhang et al., 2023), where ground truth edited images are available, we follow the original evaluation protocol by computing the metrics using these ground truth images rather than the inputs. In contrast, for Emuedit (Sheynin et al., 2024), which provides only input images, we compute the L1, and DINO metrics based on the inputs and omit the CLIP$_I$ metric.

## 4 USER-CONTROLLED EDITING: ABLATIONS AND INSIGHTS

We conduct extensive ablation studies to isolate and evaluate the individual effects of momentum, orthogonality, and normalization in both $\Delta D_t^{I,\varnothing}$ and $\Delta D_t^{T,I}$. Each hyperparameter is tested separately, with the most notable quantitative results summarized in Tab. 3 and comprehensive illustration of the corresponding visual effects are presented in Fig. 4, Appendix Fig. C.1 and Appendix Fig. C.3. Additionally, we plot the L2-norm of the denoised prediction across the denoising schedule, comparing IP2P-APG (with and without each component) to the original IP2P in Appendix Fig. B.4.

As shown in Tab. 3, we carefully select hyperparameters that strike a balance between fidelity and editing performance by setting both text ($\beta_T$) and image momentum ($\beta_I$) to small positive values. This choice maintains a robust level of text guidance without causing overly aggressive updates in the denoising process. In contrast, APG yields better FID in text-to-image generation with negative momentum (Sadat et al., 2025). However, our experiments revealed that negative text momentum, while boosting fidelity to the original content, tended to reduce the model's ability to follow editing instructions. Thus, the trade-off leaned in favor of small positive momentum values, which preserved meaningful text guidance and ensured more consistent editing outcomes. Intuitively, neg-

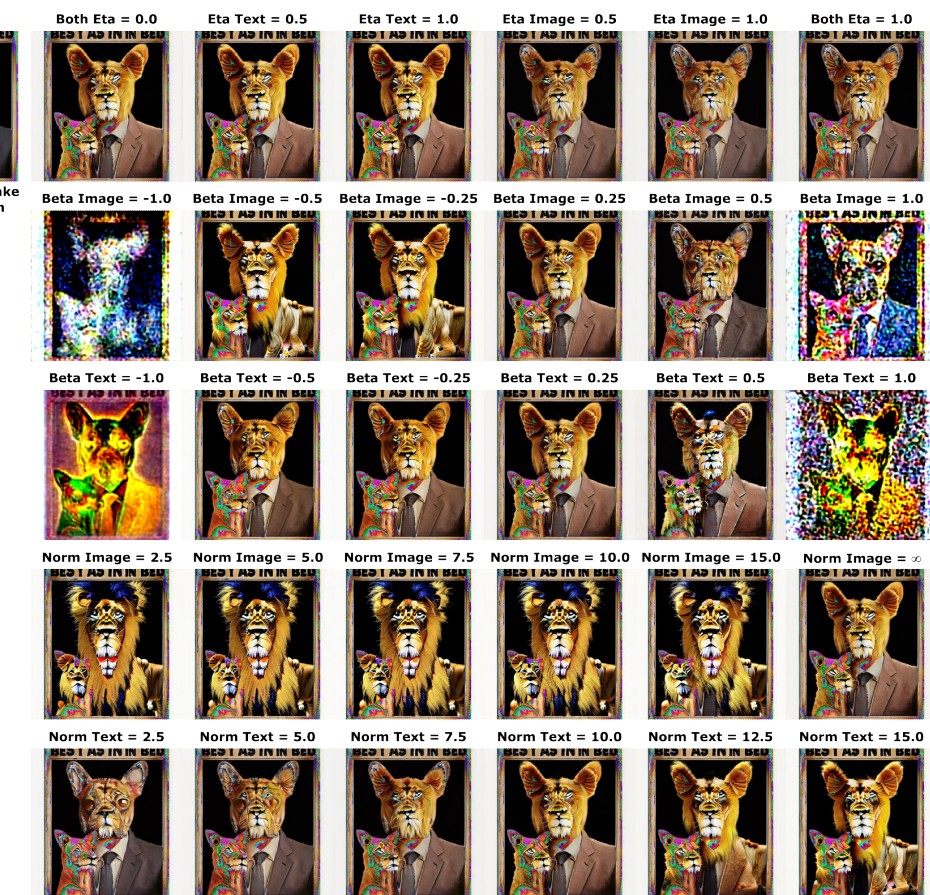

Figure 4: We demonstrate how IP2P-APG enables finer-grained, user-steerable control over the editing process by isolating the impact of each component. For clarity, symbolic notations are decoded (e.g., Eta Text refers to $\eta_T$, Norm Text refers to $r_T$ etc.). Except for the component being evaluated, all other parameters are fixed at their optimal values as specified in Tab. 3. Input image and instruction is taken from IP2P test dataset (Brooks et al., 2023) and editing performed with their SD1.5 checkpoint with ($s_T = 15.0, s_I = 1.5$).

ative momentum in text-to-image models serves a purpose in early denoising steps by exploring diverse directions in the noise space. However, in our image and text dual-conditioned framework, consecutive denoising steps should maintain a relatively consistent direction. In the second row of Fig. 4, we visualize the effect of applying momentum to the $\Delta D_t^{I,\varnothing}$: although the dog's head is successfully replaced with a lion's, negative momentum values introduce an unintended color and texture shift in the suit. The third row demonstrates the visual effect of momentum in $\Delta D_t^{T,I}$, where a positive momentum value enables the model to erase the dog's eyes more effectively.

On the other hand, normalizing $\Delta D_t^{T,I}$ proved to be essential in preventing abrupt changes during the denoising phase. By controlling the scale of these updates, normalization helped the framework to stabilize both the text and image guidance signals. At the same time, applying normalization to $\Delta D_t^{I,\varnothing}$ improved textual adherence by constraining the magnitude of image guidance; however, this came at the cost of reduced fidelity to the original content, suggesting that the primary source of unstable updates arose from the coupling of text and image guidance rather than the image component alone. The final two rows of Fig. 4 illustrate the role of image and text normalization factors ($r_I$ and $r_T$), which act as explicit regularizers by constraining the magnitude of guidance signals during denoising. Higher norm values result in stronger guidance effects, while Norm Image = $\infty$ corresponds to the absence of regularization—allowing the $\Delta D_t^{I,\varnothing}$ to influence the generation with its maximum possible strength.

Lastly, we investigated the influence of the parallel components, represented by varying $\eta$ values for both $\Delta D_t^{T,I}$ and $\Delta D_t^{I,\varnothing}$. The effect of the parallel components was comparatively minor, as be seen

Table 3: Ablation studies on MagicBrush (Zhang et al., 2023) test set with MagicBrush SD1.5 checkpoint, where $(s_T, s_I) = (15.0, 1.5)$.

| Ablation | Hyperparameters | L1 ($\downarrow$) | DINO ($\uparrow$) | CLIP$_I$($\uparrow$) | CLIP$_T$($\uparrow$) | CLIP$_{sim}$($\uparrow$) | CLIP$_{dir}$($\uparrow$) |
|---|---|---|---|---|---|---|---|
| $\eta_T, \eta_I$ | $(\eta_T, \eta_I) = (0,1)$ | 0.0626 | 0.8907 | 0.9262 | 0.2841 | 0.8748 | 0.1294 |
| | $(\eta_T, \eta_I) = (1,0)$ | 0.0669 | 0.8747 | 0.9175 | 0.2862 | 0.8555 | 0.1361 |
| $\beta_T$ | $(\beta_T, \beta_I) = (-0.25,0.25)$ | 0.0628 | 0.8842 | 0.9238 | 0.2853 | 0.8697 | 0.1317 |
| | $(\beta_T, \beta_I) = (0,0.25)$ | 0.0641 | 0.8810 | 0.9215 | 0.2857 | 0.8640 | 0.1344 |
| $r_T, r_I$ | $(r_T, r_I) = (2.5,0)$ | **0.0602** | **0.8973** | **0.9304** | 0.2805 | **0.8910** | 0.1147 |
| | $(r_T, r_I) = (7.5,7.5)$ | 0.0861 | 0.8123 | 0.8866 | **0.2882** | 0.8068 | 0.1446 |
| | $(r_T, r_I) = (0,2.5)$ | 0.1245 | 0.7285 | 0.8455 | 0.2868 | 0.7583 | **0.1485** |
| | $(r_T, r_I) = (0,10.0)$ | 0.1228 | 0.7321 | 0.8476 | 0.2869 | 0.7616 | 0.1484 |
| Proposed | $(\beta_T, \beta_I) = (0.25,0.25)$ $(r_T, r_I) = (10.0,0)$ $(\eta_T, \eta_I) = (0,0)$ | 0.0668 | 0.8754 | 0.9175 | 0.2859 | 0.8556 | 0.1366 |

in Tab. 3, especially when compared to the more dominant impact of momentum and normalization choices. Consequently, to simplify our approach without sacrificing performance, we set both $\eta_T$ and $\eta_I$ parameters to zero. Visually in Fig. 4, as shown in the first row, the parallel component in $\Delta D_t^{T,I}$ has minimal impact, while the parallel component in $\Delta D_t^{I,\varnothing}$ leads to undesired retention of original content—for example, the dog's eyes remain visible.

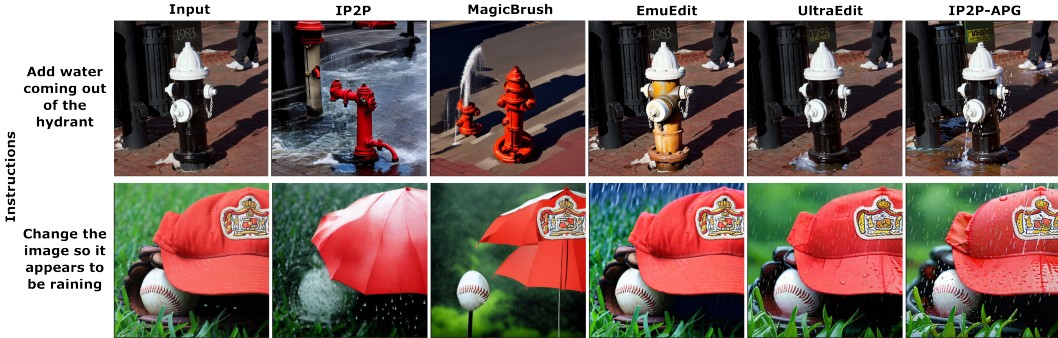

Figure 5: Comparison of methods using IP2P-APG with $(s_T, s_I) = (15.0, 1.5)$ and the UltraEdit (Zhao et al., 2025) SD3 checkpoint on the EmuEdit test dataset (Sheynin et al., 2024). As observed, IP2P-APG successfully applies the required edits in both cases while preserving the overall structure and retaining most of the original details.

## 5 CONCLUSION AND LIMITATIONS

In this work, we introduced IP2P-APG, a plug-and-play framework that integrates adaptive projection strategies into instruction-guided image editing to improve the balance between instruction adherence and image fidelity. By decomposing guidance signals into orthogonal components and incorporating momentum and normalization terms, our method extends the effective range of guidance scales and provides users finer, more predictable control over each aspect of the editing process.

Our analysis clarifies the denoising dynamics under joint image and text conditioning and shows how momentum, normalization, and parallel components shape the edit trajectory. While our approach improves fidelity and prompt adherence without retraining by relying on the base model's original weights, the ultimate performance remains bounded by the capacity of the underlying architecture.

## REPRODUCIBILITY STATEMENT

Our method is training-free: we apply Eq. (14) on top of publicly available Stable Diffusion checkpoints from the InstructPix2Pix (Brooks et al., 2023), MagicBrush (Zhang et al., 2023), and UltraEdit (Zhao et al., 2025) codebases. All datasets are public (EmuEdit (Sheynin et al., 2024), MagicBrush (Zhang et al., 2023), InstructPix2Pix (Brooks et al., 2023)). Because no fine-tuning is required, the algorithm is a lightweight wrapper around the standard sampler and can be implemented in a few lines of Python. The exact hyperparameters used in our experiments are listed in Tab. 3.

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

## USE OF LARGE LANGUAGE MODELS (LLMS)

We have used LLMs only to polish writing in this paper.

## A  RELATED WORKS

### TEXT-GUIDED IMAGE TRANSLATION

A well-established approach to image translation leverages pre-trained diffusion models. The process involves first inverting the input image into a latent representation, then refining it through text-based prompts. SDE-Edit (Meng et al., 2021) employs a stochastic perturbation process to first introduce controlled noise into images, followed by a guided denoising step conditioned on text prompts to achieve editing, Null-Text Inversion (Mokady et al., 2023) enables edits on real images by inverting an input image using a null-text embedding, while EDICT (Wallace et al., 2023) introduces a dual noise-vector inversion method to improve image reconstruction and alignment with textual prompts. Imagic (Kawar et al., 2023) fine-tunes the diffusion model itself to better interpret complex text instructions.

Other methods focus on manipulating attention and spatial features to guide edits more effectively. Prompt-to-Prompt (P2P) (Hertz et al., 2022) modifies attention maps by injecting those from an input caption into the target caption, whereas Plug-and-Play (PNP) (Tumanyan et al., 2023) enhances editing precision by integrating spatial features alongside attention maps. Another class of models incorporates masks as additional inputs to improve localized editing. Imagen Editor (Wang et al., 2023) and SmartBrush (Xie et al., 2023) extend text-to-image models by conditioning them on both the input image and a corresponding mask.

Despite these advancements, text-based image translation methods often suffer from inconsistencies and require additional inputs, such as highly detailed textual descriptions of both the input and target images or explicitly defined masks, making them less flexible for general use.

### INSTRUCTION-GUIDED IMAGE EDITING

Following InstructPix2Pix (Brooks et al., 2023), several methods—including MagicBrush (Zhang et al., 2023), UltraEdit (Zhao et al., 2025), and EmuEdit (Sheynin et al., 2024)—have been developed to address its limitations, primarily by improving dataset quality. Since InstructPix2Pix is trained exclusively on synthetic data, which can introduce noise and lack the diversity of real-world images, these approaches aim to mitigate performance bottlenecks caused by suboptimal training samples. However, while these methods focus on enhancing data quality, they do not fundamentally change inference process of InstructPix2Pix, setting our approach apart.

More recently, additional techniques have emerged to improve fidelity in instruction-guided image editing. UIP2P (Simsar et al., 2024) incorporates cycle-consistency loss (Zhu et al., 2017) to reduce dependency on paired datasets, while SeedEdit (Shi et al., 2024) introduces a diffusion-based framework that unifies generation and editing, striving to balance both tasks effectively. While these methods offer a better fidelity they differ from our training-free approach.

## B  ADDITIONAL QUANTITATIVE RESULTS

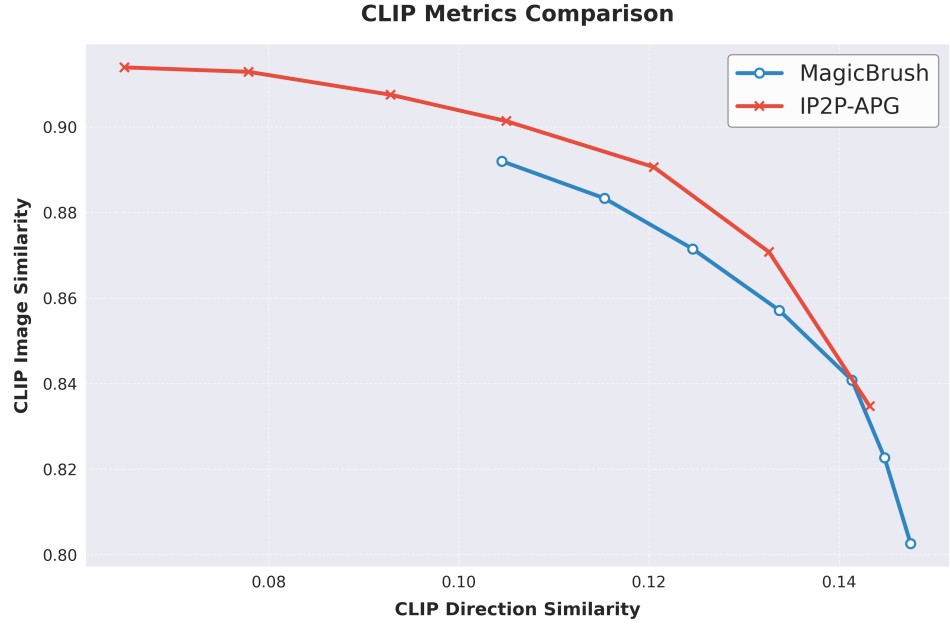

Figure B.1: Comparison of CLIP Metrics on the InstructPix2Pix (Brooks et al., 2023) test set for IP2P-APG and the original IP2P, using the IP2P checkpoint with Stable Diffusion v1.5 ($s_T = 7.5$ with varying $s_I \in [1.0, 2.2]$).

Figure B.2: Comparison of CLIP Metrics on MagicBrush (Zhang et al., 2023) test set for IP2P-APG and MagicBrush, using their fine-tuned checkpoint with Stable Diffusion v1.5. ($s_T = 7.5$ with varying $s_I \in [1.0, 2.2]$)

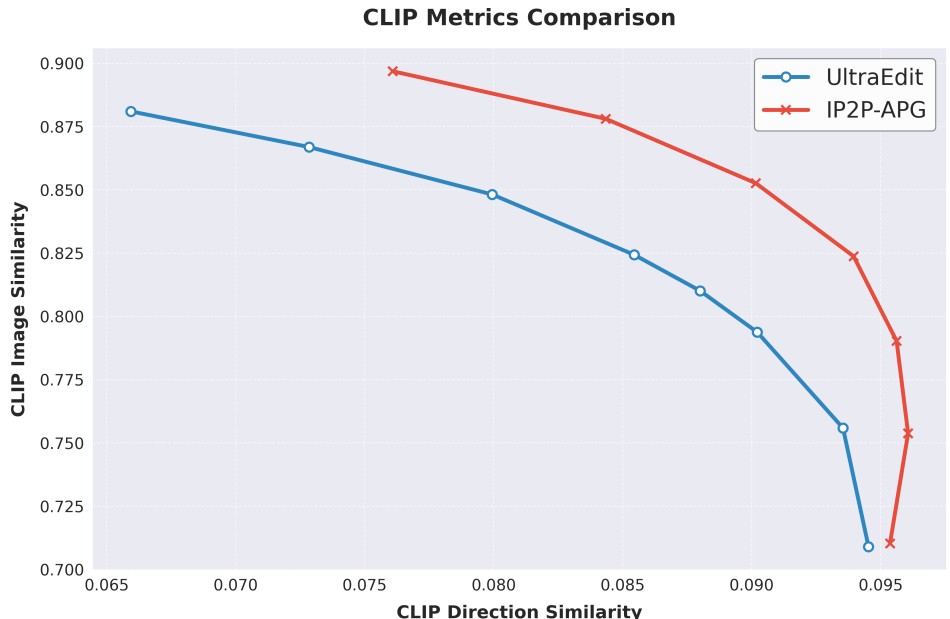

Figure B.3: Comparison of CLIP Metrics on the EMUEDIT (Sheynin et al., 2024) test set for IP2P-APG and UltraEdit, using the Stable Diffusion 3 checkpoint from (Zhao et al., 2025). ($s_T = 7.5$ with varying $s_I \in [1.0, 2.2]$)

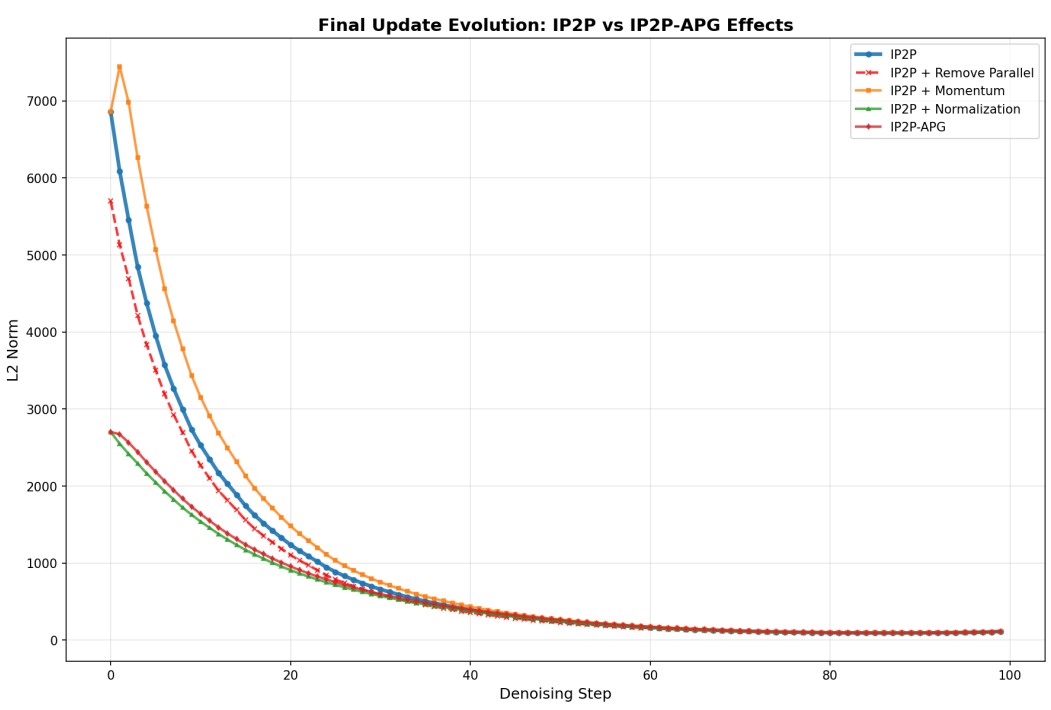

Figure B.4: Evolution of the denoised estimate (predicted clean latent) magnitude throughout denoising, contrasting IP2P with IP2P-APG. The value at denoising step $t$ is $||\tilde{D}_\theta(z_t, c_I, c_T)||_2$, averaged over 100 test images from MagicBrush. Normalization in $\Delta D_t^{T,I}$ decouples direction from scale, enforcing a bounded energy on the denoised prediction and yielding smoother, lower-norm trajectories.

## C ADDITIONAL QUALITATIVE RESULTS

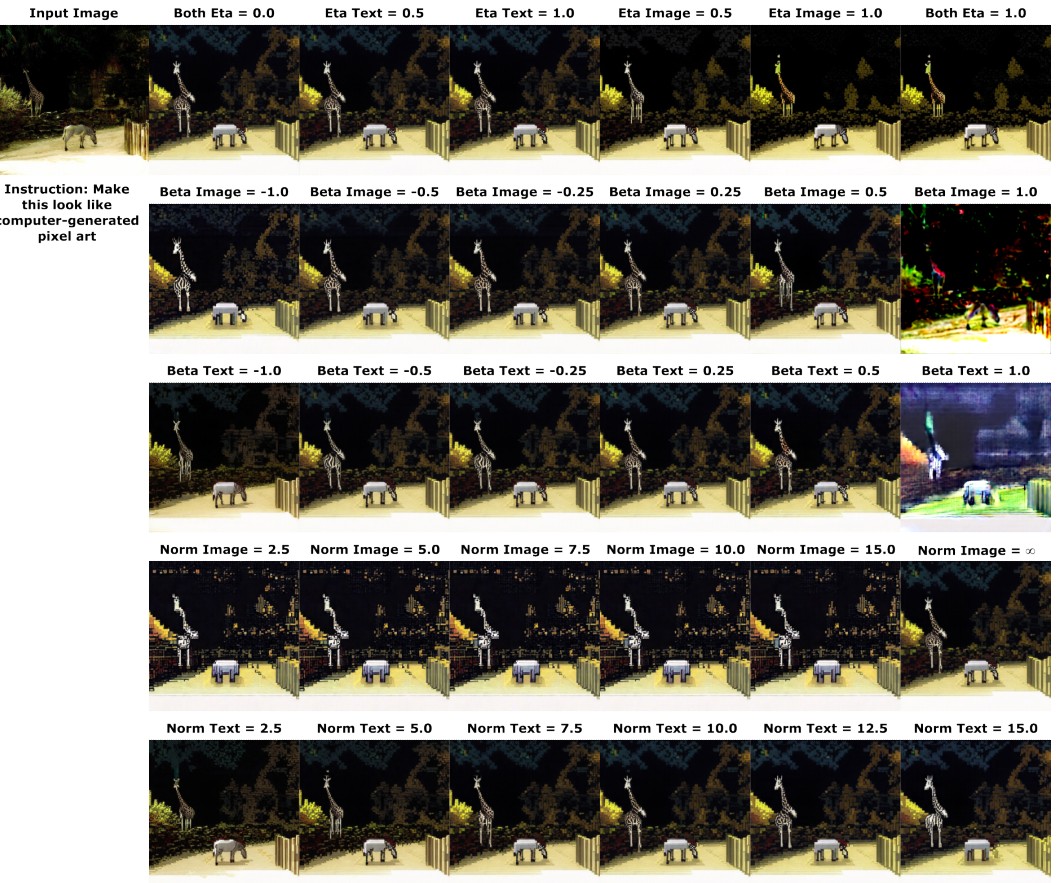

Figure C.1: Component-wise analysis, analogous to Fig. 4, on an EmuEdit (Sheynin et al., 2024) example edited with UltraEdit's SD3 checkpoint (Zhao et al., 2025) ($s_T = 7.5, s_I = 1.5$).

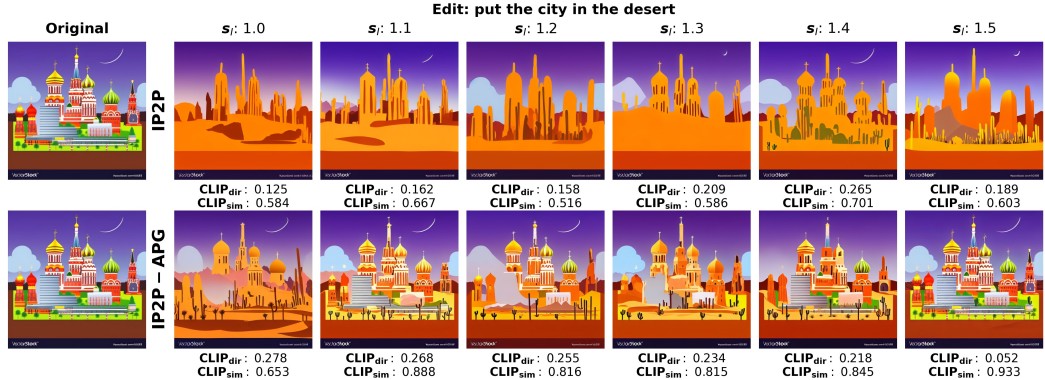

Figure C.2: Comparison of IP2P and IP2P-APG at $s_T = 7.5$ with varying $s_I$ values. Using identical model weights, IP2P-APG yields significantly improved fidelity, better preserving building structures and maintaining a more accurate color palette while consistently yielding higher CLIP scores.

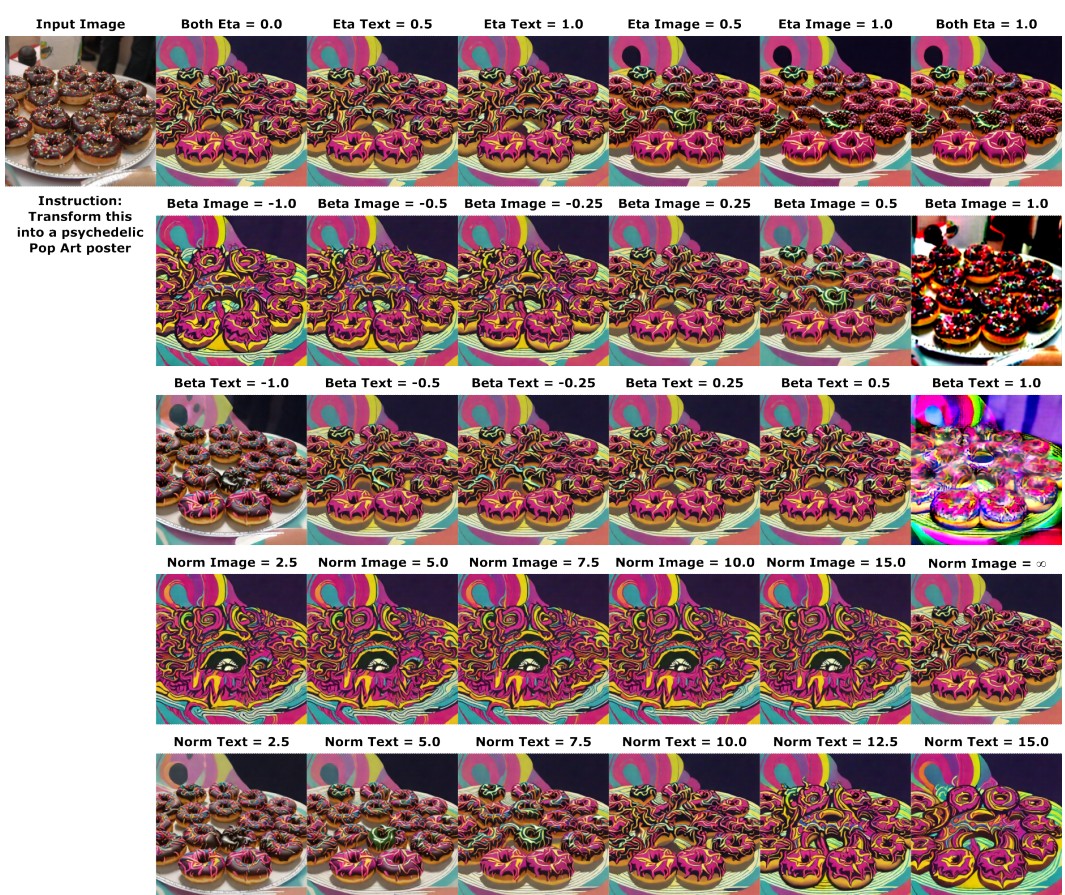

Figure C.3: Component-wise analysis, analogous to Fig. 4 and Fig. C.1, on an EmuEdit (Sheynin et al., 2024) example edited with UltraEdit's SD3 checkpoint (Zhao et al., 2025) ($s_T = 7.5, s_I = 1.5$).

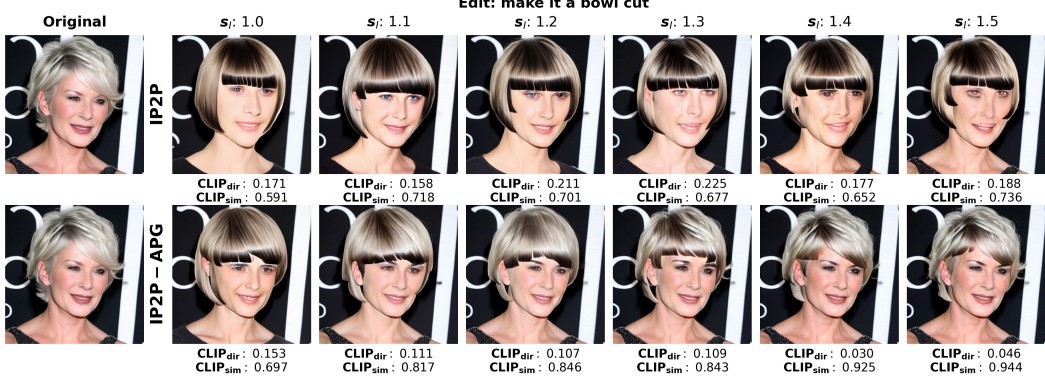

Figure C.4: Comparison of IP2P and IP2P-APG at $s_T = 7.5$ with varying $s_I$ values. With identical model weights, IP2P-APG achieves notable improvements in both fidelity and textual adherence, performing accurate and targeted edits. It preserves the woman's identity while modifying only the hairstyle as instructed, whereas IP2P alters her facial features as well.

