# OpenReview forum: "Adaptive Projected Guidance for Controllable Instruction-Guided Image Editing"
_ICLR.cc/2026/Conference — Submitted to ICLR 2026_

### Official Review · Reviewer_tje8 · 2025-10-30

**Soundness:** 2
**Presentation:** 2
**Contribution:** 2
**Rating:** 2
**Confidence:** 4

**Summary:**

This paper proposes to replace the conventional CFG with adaptive projected guidance, therefore mitigating the inherent limitations of CFG at high guidance scales. It expands the controllable parameters space, and the editing performance is improved across different diffusion backbones according to the quantitative results.

**Strengths:**

1. It adopts the adaptive guidance into the IP2P framework, which can enable high guidance scale for text condition. The proposed method can better preserve the original layout compared to the original IP2P.

2. The idea is straightforward and intuitive.

**Weaknesses:**

1. I consider this work as a simple combination of the existing adaptive projected guidance and the IP2P framework. Although it can improve the performance of original IP2P, there is not enough technical novelty or new insight brought by the paper itself. This is the fundamental limitation of this paper as I see it.

2. The editing results showcased by the paper are not strong enough. The example in the teaser under s_i=1.4 does not look good enough. The two examples in Figure 3 are also not competitive enough compared to the baselines. I would also suggest the authors to look for more visually pleasing [input image + instruction] examples to replace several existing ones shown in the paper. Input images with cleaner background and more obvious editing changes are recommended.

**Questions:**

It looks like that the images in Figure 4 are not scaled to the correct range to visualize, so they have the distorted overflowed colors. Could the authors check the images again?

---

### Official Review · Reviewer_PsA3 · 2025-10-31

**Soundness:** 3
**Presentation:** 2
**Contribution:** 2
**Rating:** 4
**Confidence:** 3

**Summary:**

To address the imbalanced edits on InstructPix2Pix (IP2P) applying separate classifier-free guidance (CFG), the paper present IP2P-APG, a plug-and-play extension of IP2P that repurposes APG to improve the balance between instruction adherence and content preservation in image editing tasks. It expands the controllable parameter space, allowing users to have more precise control over the editing process.

**Strengths:**

- This paper is of high practical value. The paper repurposes APG to improve the balance between instruction adherence and content preservation in image editing tasks.
- This paper designs a rich set of experiments to demonstrate the advantages of the method, and compares the effects of different hyperparameters.

**Weaknesses:**

- The contribution lacks novelty. The proposed method builds entirely upon the existing InstructPix2Pix (IP2P) and Adaptive Projected Guidance (APG) frameworks, amounting to a direct substitution of APG for the original classifier-free guidance component in IP2P without introducing any additional conceptual or algorithmic innovations.
- The paper exhibits a substantive content deficit. Beyond the experimental section, its technical work is confined to Subsection 2.4, where it is further diluted by an excessive number of definitional equations that add little interpretive value. Overall, the paper offers an unduly prolix account of prior work and presents its key derivations with unduly prolix detail, raising the suspicion of padding.

**Questions:**

- Could the authors please clarify the theoretical innovations that may have been missed by me, especially principles or derivations distinct from the original Adaptive Projected Guidance formulation?
- It is recommended that the paper be restructured to allocate substantially more space to a detailed exposition of the present work. Prior-work summaries should be condensed so that the core methodological design, theoretical justification, and ablative analyses receive adequate emphasis, thereby enabling readers to assess the contribution without wading through disproportionately lengthy background sections.

---

### Official Review · Reviewer_SNT7 · 2025-11-01

**Soundness:** 3
**Presentation:** 3
**Contribution:** 2
**Rating:** 4
**Confidence:** 5

**Summary:**

This paper proposes IP2P-APG, a training-free extension to InstructPix2Pix (IP2P) that improves controllability in instruction-guided image editing by incorporating Adaptive Projected Guidance (APG). The key idea is to decompose the text and image guidance signals into parallel and orthogonal components, and regulate them separately using momentum, normalization, and projection, which allows users to adjust how strongly the model follows the instruction versus how much it preserves the original image. The proposed method is plug-and-play, requires no retraining, and works across various diffusion backbones Experiments show that IP2P-APG consistently yields promising editing results.

**Strengths:**

S1. Quantitative results in the MagicBrush and Emuedit test set are promising, and authors have reported extensive experimental results using both DiT-based architecture (SD3) and U-Net based architecture (SD 1.5).

S2. The authors clearly verified the effects of each component (in terms of hyperparameter isolation) both qualitatively (Fig. 4, Fig C.1. - C.4.) and quantitatively (Table 3). It strengths the proposed thesis’ requiredness.

**Weaknesses:**

W1. The novelty of the proposed method is limited. The authors proposed the IP2P-APG framework, which combines (the tailored version of) adaptive projected guidance (APG) with instructpix2pix (IP2P). However, two components (IP2P, APG) are already defined and verified in the previous works, and I think it’s hard to find novel components and core contributions within the proposed method. This is my major concern with this paper. If there are any other novel contributions, please emphasize it; otherwise, please re-organize the method section to strongly support the paper’s contributions.

W2. The experiment section only focuses on the object change and addition tasks, such as “Add kids playing …”. I was wondering if the method is applicable to more difficult tasks, such as 1) object duplication, 2) object deletion, and 3) enlarging or shrinking the object size. Extensive experiments with additional tasks is required.

W3. As mentioned in S2, authors provided brief analysis on hyperparameters and each proposed component. I saw that the proposed method contains (at least) 8 hyperparameters ($\beta_T$, $\beta_I$, $r_T$, $r_I$, $\eta_T$, $\eta_I$, $s_T$, $s_I$), and the sensitivity of each hyperparameters should be also analyzed. I was wondering how the result is quantitatively affected by the variation of each hyperparameter, beyond ablation. In addition, from the last column of Figure 4, I think the proposed method is somewhat sensitive to $\beta$ hyperparameter. Is there any plausible strategy to manually tune these hyperparameters?

W4. More qualitative comparison with baseline is required. The main paper shows only four examples (Included in Figure 3 and 5), and more qualitative results that emphasize the strengths and effectiveness of the proposed IP2P-APG across baselines is required.

W5. The authors reported ablation study results using SD v1.5 checkpoint. Could authors kindly provide the ablation study results with SD v3 (*i.e.* Transformer-based diffusion model) checkpoint?

**Questions:**

Please check the weakness section.

---

### Official Review · Reviewer_LDmd · 2025-11-01

**Soundness:** 2
**Presentation:** 3
**Contribution:** 1
**Rating:** 2
**Confidence:** 4

**Summary:**

The paper applies Adaptive Projected Guidance (APG) to intruction-guided diffusion model (IP2P) as a plug-and-play framework. The model-agnostic framework shows improvement across benchmarks.

**Strengths:**

- The work shows that APG works on editing model such as IP2P

**Weaknesses:**

- Lack of technical novelty since the APG paper already shows generalization across image generation models.
- The method seems fragile qualitatively in Figure 1 as $s_I$ works at 1.1/1.5 but does not work at other values.

**Questions:**

- Is there technical difference in applying APG to image generation models vs image editing models.

---

### Author Response · Authors · 2025-11-13
**Clarifying Our Contribution: Controlling Instruction-Guided Editing**

We thank the reviewers for their constructive comments and acknowledge the concern regarding the novelty of our contribution. We apologize for any misunderstanding about the intended scope of our work, and we appreciate the opportunity to clarify this in our rebuttal.

Our work does not propose a new image editing paradigm, nor do we claim to introduce a fundamentally new architectural innovation. As stated explicitly in both the abstract and the main paper, our contribution lies in repurposing Adaptive Projected Guidance (APG), a technique originally designed to mitigate oversaturation at high classifier-free guidance scales in text-to-image or class-conditional diffusion models, for a completely different and previously unexplored purpose. In the original APG formulation, normalization, orthogonal projection, and reverse momentum are all introduced to achieve a single unifying goal: improve sampling quality by reducing saturation and stabilizing classifier-free guided denoising. In contrast, our work reinterprets these same mechanisms as independent control handles that shape the editing trajectory itself, rather than serving purely as corrective terms. Our intention is to get more out of already pretrained instruction-guided image editing models, which are normally much harder to control, by enabling richer steering without modifying or retraining them.

More specifically, the original InstructPix2Pix framework exposes only two global guidance strength parameters, which significantly limits how users can influence the editing process. Our IP2P-APG formulation expands this space of controllability by allowing users to adjust the norm of the guidance signal originating from either the text prompt or the input image at every diffusion step, regulate the momentum with which these signals accumulate through the denoising trajectory, and control the orthogonality between text-based and image-based guidance so that these influences can be blended, separated, or emphasized as desired. Unlike APG’s original usage, where all components jointly serve to reduce oversaturation, here they act as expressive levers that determine how the model interprets and executes semantic instructions. We also note that IP2P-APG allows the use of higher guidance scales without overemphasizing a direction and degrading content fidelity, which is consistent with APG’s original benefit and serves as an empirically validated advantage over existing methods, although our central contribution remains the expanded user control enabled by these mechanisms as stated in the title of our paper.

It is also important to emphasize that our goal is not to create a new editing tool aimed at improving benchmark performance. We do report competitive scores, but these are ultimately limited by the underlying generative model as stated in the limitations. IP2P-APG exposes fine-grained controls to the user, so different images naturally require different parameter settings. A single configuration that works equally well for all inputs is unrealistic and not the aim of IP2P-APG. The purpose of our method is to give users meaningful control over how edits evolve, rather than fix the behavior to one universal choice.

As we empirically demonstrate in Figure 4, Figure C.1, and Figure C.3, manipulating these aspects leads to a wide spectrum of editing behaviors. Users can make subtle refinements or apply substantial transformations depending on how guidance norm, momentum, and orthogonality are configured. Even when the model, the prompt, and the input image remain the same, different configurations of these parameters produce clearly distinct final images. This demonstrates that IP2P-APG provides a level of controllability that is absent in the original InstructPix2Pix pipeline.

In summary, our work presents a novel application and functional reinterpretation of an existing guidance mechanism, not a new generative framework. The contribution lies in identifying that the internal components of APG can be repurposed as intuitive user-facing controls for instruction-guided image editing, and in demonstrating their practical effectiveness across diverse editing scenarios.

---

### Meta-Review · Area_Chair_PF6e · 2026-01-03

**Summary:**

The paper was reviewed by 4 experts obtaining scores 4224. The major concerns are listed in the next field. The incremental novelty of the paper was a major concern, which was not addressed convincingly. In addition the experiments were not fully convincing. Thus, it is unlikely that the response would have assuaged the reviewers concerns.

**Reviewer Concerns:**

All reviewers raised a concern about novelty: the proposed work combines the existing APG with IP2P, and thus provides no new theoretical, algorithmic, or architectural contributions. In the response, authors state that their goal is to reinterpret the mechanisms in APG as controls for changing the edit trajectory.

Other issues indicate that the experiments were not entirely convincing:
- experiments on more difficult tasks
- ablation study on hyperparameters
- needs more qualitative comparisons
- needs more convincing results.

**Reviewer Scores:**

Given the novelty issues, the AC thinks that the reviewers would not have changed their scores above marginal reject.

---

### Decision · Program_Chairs · 2026-01-26

Reject